

# Primary emissions of glyoxal and methylglyoxal from laboratory measurements of open biomass burning

Kyle J. Zarzana[1,2,*], Vanessa Selimovic[3], Abigail R. Koss[1,2,4,**], Kanako Sekimoto[1,2,5], Matthew M. Coggon[1,2], Bin Yuan[1,2,***], William P. Dubé[1,2], Robert J. Yokelson[3], Carsten Warneke[1,2], Joost A. de Gouw[1,2,4], James M. Roberts[1], and Steven S. Brown[1,4]

[1]NOAA Earth System Research Laboratory (ESRL) Chemical Sciences Division, Boulder, CO 80305, USA
[2]Cooperative Institute for Research in Environmental Sciences, University of Colorado Boulder, Boulder, CO 80309, USA
[3]Department of Chemistry and Biochemistry, University of Montana, Missoula, MT 59812, USA
[4]Department of Chemistry and Biochemistry, University of Colorado Boulder, Boulder, CO 80309, USA
[5]Graduate School of Nanobioscience, Yokohama City University, Yokohama, Kanagawa 236-0027, Japan
[*]now at: Department of Chemistry and Biochemistry, University of Colorado Boulder, Boulder, CO 80309, USA
[**]now at: Department of Civil and Environmental Engineering, Massachusetts Institute of Technology, Cambridge, MA 02142, USA
[***]now at: Institute of Environment and Climate Research, Jinan University, Guangzhou 510632, China

**Correspondence:** S. S. Brown (Steven.S.Brown@noaa.gov)

**Abstract.** We report the emissions of glyoxal and methylglyoxal from the open burning of biomass during the NOAA-led 2016 FIREX intensive at the Fire Sciences Laboratory in Missoula, MT. Both compounds were measured using cavity enhanced spectroscopy, which is both more sensitive and more selective than methods previously used to determine emissions of these two compounds. A total of 75 burns were conducted, using 33 different fuels in 8 different categories, providing a far more

comprehensive dataset for emissions than was previously available. Measurements of methylglyoxal using our instrument suffer from spectral interferences from several other species, but methylglyoxal emissions could be constrained within roughly a factor of 2. Methylglyoxal emissions were 2-3 times higher than glyoxal emissions on a molar basis, in contrast to previous studies that report methylglyoxal emissions lower than glyoxal emissions. Methylglyoxal emission ratios for all fuels averaged 3.6±2.4 ppbv methylglyoxal/ppmv CO, while emission factors averaged 0.66±0.50 g methylglyoxal/kg fuel burned. Primary

emissions of glyoxal from biomass burning were much lower than previous laboratory measurements, but consistent with recent measurements from aircraft. Glyoxal emission ratios for all fuels averaged 1.4±0.7 ppbv glyoxal/ppmv CO, while emission factors averaged 0.20±0.12 g glyoxal/kg fuel burned, values that are at least a factor of 4 lower than assumed in previous estimates of the global glyoxal budget. While there was significant variability in the glyoxal emission ratios and factors between the different fuel groups, glyoxal and formaldehyde were highly correlated during the course of any given

fire, and the ratio of glyoxal to formaldehyde, $R_{GF}$, was consistent across many different fuel types, with an average value of 0.068±0.018. While $R_{GF}$ values for fresh emissions were consistent across many fuel types, further work is required to determine how this value changes as the emissions age.



*Copyright statement.* TEXT

# 1 Introduction

In addition to the large primary emissions of gases and particulate matter, the secondary chemistry that occurs downwind of fires can play an important role in numerous atmospheric processes. Ozone ($O_3$), peroxy nitrates such as acetyl peroxynitrate (PAN), and organic aerosol are frequently enhanced in downwind fire plumes (e.g. Yokelson et al., 2009; Akagi et al., 2012; Alvarado et al., 2015; Liu et al., 2016), and in urban areas influenced by biomass burning, emissions from fires have been shown to increase $O_3$ above the 70 ppbv standard set by the EPA (Brey and Fischer, 2016; Gong et al., 2017). Modeling of the chemistry of biomass burning plumes has found that carbonyls such as formaldehyde, methylglyoxal, and 2,3-butanedione play a large role in the formation of both $O_3$ and PAN (Mason et al., 2001; Müller et al., 2016), either through reactions with hydroxyl radicals or photolysis. Carbonyl photolysis leading to $O_3$ production has also been observed in other regions, such as oil and natural gas producing basins (Edwards et al., 2014). In addition to contributing to $O_3$ formation, photolysis of carbonyls such as acetone and methylglyoxal can lead to the formation of PAN (Fischer et al., 2014; Müller et al., 2016). Understanding the impact of carbonyls on fire plume chemistry requires accurate measurements of emissions of these compounds, but those data are lacking for several carbonyl species, particularly small $\alpha$-dicarbonyls such as glyoxal and methylglyoxal.

Along with glyoxal and methylglyoxal, numerous other carbonyl species such as formaldehyde have been detected in fire plumes (e.g. Akagi et al., 2011; Stockwell et al., 2015; Koss et al., 2018). While methylglyoxal's absorption cross section is relatively weak and unstructured, the cross sections of glyoxal and formaldehyde in the visible and ultraviolet respectively are large and structured, enabling the detection of those two molecules from space using remote sensing instruments such as the Scanning Imaging Absorption Spectrometer for Atmospheric Cartography (SCIAMACHY, Wittrock et al., 2006; Myriokefalitakis et al., 2008), the Global Ozone Monitoring Experiment-2 (GOME-2, Lerot et al., 2010), the Ozone Monitoring Instrument (OMI, Alvarado et al., 2014; Chan Miller et al., 2014) or the Tropospheric emissions: Monitoring pollution instrument (TEMPO, Zoogman et al., 2017).

The column abundances of these two molecules are enhanced in regions influenced by biomass burning (Chan Miller et al., 2014), but the main source of both molecules globally is oxidation of larger volatile organic compounds (VOCs) (Shim et al., 2005; Fu et al., 2008; Fortems-Cheiney et al., 2012). The relative yields of glyoxal and formaldehyde depend in part on the precursor VOC, and the ratio of glyoxal to formaldehyde, $R_{GF}$, is higher in regions dominated by emissions of aromatic VOCs than it is in regions dominated by emissions of isoprene (Chan Miller et al., 2016; Kaiser et al., 2015). $R_{GF}$ has been proposed as a metric for examining VOC chemistry from space (Vrekoussis et al., 2010; Chan Miller et al., 2014; Kaiser et al., 2015), as glyoxal and formaldehyde have similar atmospheric lifetimes with respect to photolysis and OH ~3 hours), but have different yields from VOC oxidation. However, doing so requires both accurate yields from oxidation reactions and a thorough understanding of direct emissions from sources producing both compounds, such as biomass burning.

Together, direct emissions from biomass burning and biofuel (biomass used as an energy source) have been estimated to contribute 20% of the glyoxal budget, but only 3.5% of the methylglyoxal budget (Fu et al., 2008; Stavrakou et al., 2009a).





While there have been numerous measurements of formaldehyde emissions from biomass burning both in the laboratory and the field, glyoxal and methylglyoxal emissions in current models are based on only two laboratory studies (McDonald et al., 2000; Hays et al., 2002). These studies examined only a limited number of fuels, and the method used in those studies to quantify carbonyl emissions is now known to be prone to interferences (Karst et al., 1993; Achatz et al., 1999). The reported emissions

of glyoxal and methylglyoxal from those studies are contradicted by field measurements from aircraft that find significantly less glyoxal but more methylglyoxal in fresh biomass burning plumes than was measured in the lab (Zarzana et al., 2017). Additionally, the laboratory studies reported that glyoxal and formaldehyde are emitted at a molar ratio of 1, roughly an order of magnitude higher than what was observed in the field (Zarzana et al., 2017) and from remote sensing platforms over regions dominated by biomass burning (Chan Miller et al., 2014; Stavrakou et al., 2016).

Models have generally been able to reproduce the formaldehyde columns observed by satellites (Stavrakou et al., 2009b; Boeke et al., 2011), but have had varying success reproducing glyoxal columns. Several studies comparing model outputs to satellite columns retrieved by SCIAMACHY and GOME-2 have found that the models underestimate global glyoxal emissions (Myriokefalitakis et al., 2008; Stavrakou et al., 2009a; Lerot et al., 2010). A more recent study by Stavrakou et al. (2016) examined emissions from crop residue fires in the north China plain using data from OMI. The column $R_{GF}$ measured by OMI

($\sim$0.04-0.05) was comparable to the $R_{GF}$ values observed by Zarzana et al. (2017), and a model was able to reproduce the measured formaldehyde columns and the glyoxal enhancements observed during the height of the burning season. However, Stavrakou et al. (2016) used glyoxal emissions from the two previous laboratory studies, which are both higher than recent field data and imply that $R_{GF}$ should be close to 1. Better measurements of emissions of glyoxal and methylglyoxal from biomass burning from a wider range of fuels, and subsequent chemistry following emission, are needed to resolve these discrepancies

and provide better inputs to models.

In this work we use cavity enhanced spectroscopy (CES) to measure primary emissions of glyoxal and methylglyoxal from open burns conducted in a laboratory setting. These experiments were conducted as part of the NOAA-led Fire Influence on Regional and Global Environments Experiment (FIREX), which took place from October to November 2016 at the US Forest Service Fire Sciences Laboratory (FSL) in Missoula, MT. CES measurements of glyoxal and methylglyoxal are faster, more

sensitive, and more specific than the methods used in previous studies. Over thirty different fuel types were burned during the 2016 FIREX campaign, and, combined with the other instrumentation deployed at the FSL, our data provide the most detailed look to date at direct emissions of glyoxal and methylglyoxal from biomass burning.

## 2 Methods

### 2.1 FSL facility

Burns were conducted at the FSL during the 2016 FIREX intensive (https://www.esrl.noaa.gov/csd/projects/firex/firelab/). Details on the FSL facility (Christian et al., 2003, 2004; Burling et al., 2010) and the FIREX campaign (Selimovic et al., 2018) are given elsewhere. The data presented here were collected during the 75 stack burns conducted during the first three weeks of the campaign, and primarily come from three instruments: the NOAA Airborne Cavity Enhanced Spectrometer (ACES); the



NOAA proton-transfer-reaction time-of-flight mass spectrometer (PTR-ToF); and the University of Montana open path Fourier transform infrared spectrometer (OP-FTIR). The setup of the three instruments during the stack burns is shown in Fig. 1. The fuel bed is located in the center of the burn chamber, which during burns was pressurized to push smoke out the ceiling through a 1.6 m diameter stack past a sampling platform 17 m above the fuel bed. The flow through the stack was well mixed, with a

residence time of roughly 5 seconds. All three instruments had sampling ports on the platform, though the PTR-ToF was not mounted on the platform itself.

A total of 33 different fuels were used, including numerous burns of coniferous fuels and chaparral species. For the conifers, burns were conducted either using only one component (e.g. litter, canopy, etc.) or with realistic mixes of several components. A full list of fuels in given in the supplement (Table S1) and in Selimovic et al. (2018).

## 2.2   Instruments used

All the instruments used here have been described previously, so only brief descriptions will be provided. The species-specific uncertainties for each instrument are given in Table 1. Concentrations of all species were significantly higher than the instrument detection limits, with concentrations of glyoxal and formaldehyde during the fires ranging from 10 to either 600 (glyoxal) or 5000 ppbv (formaldehyde), and concentrations of carbon monoxide at the peak of the fire exceeded 100 ppmv.

### 2.2.1   ACES

Glyoxal and methylglyoxal were measured using the ACES instrument (Min et al., 2016). Light from an LED with a center wavelength of 455 nm was introduced into a 45 cm long cavity capped with highly reflective (R>0.99995 at 455 nm) mirrors, enabling the light to make multiple passes and resulting in an effective pathlength of 10-12 km. The light exiting the cavity entered a grating spectrometer and was imaged onto a charge-coupled device (CCD) array. The overlap between the mirror

reflectivity and the LED output resulted in a useful spectral range between 438 and 468 nm. The measured CCD counts were then converted into extinction (the sum of scattering and absorption) (Washenfelder et al., 2008). The wavelength-dependent extinction, $\alpha(\lambda)$, is due to absorption and Rayleigh scattering by gas-phase molecules and scattering and absorption by aerosol particles. The particles were removed with a filter (see below), and Rayleigh scattering was accounted for by measuring the number density in the cell. The measured extinction is then

$$\alpha(\lambda)_{measured} = \sum_{i=1}^{n} \sigma(\lambda)_i N_i \tag{1}$$

where $\sigma(\lambda)_i$ and $N_i$ are respectively the absorption cross section and number density of a given species. The measured spectra were fit using the DOAS fitting routines in the DOASIS software package (Platt and Stutz, 2008; Kraus, 2006), which took as inputs the absorption cross sections of the species of interest convolved to the resolution of the instrument (here, nitrogen dioxide ($NO_2$), glyoxal, and methylglyoxal). For each spectrum, DOASIS then determined the number density for each species

that resulted in the best agreement between the measured and calculated spectrum. Data for ACES are reported at 1 Hz.

ACES has a second channel centered at 375 nm measuring nitrous acid (HONO) and $NO_2$ that is imaged onto the CCD using the same spectrometer. Imaging two channels separated by 80 nm with the same spectrometer requires a relatively coarse





grating, resulting in a resolution for both channels of ∼1 nm full width half maximum (FWHM). Even at this resolution, the glyoxal cross section in the ACES retrieval window is highly structured and distinct from the cross sections of other molecules absorbing in the same region such as $NO_2$ and methylglyoxal. This method therefore provides a robust and direct measurement of glyoxal with a minimal need for corrections. The methylglyoxal cross section is less structured than the glyoxal cross section,

and at our resolution suffers from spectral interferences from other substituted $\alpha$-dicarbonyls such as 2,3-butanedione and 2,3-pentanedione, which have cross sections with similar structure but lower intensity.

ACES was installed on the platform (see Fig. 1) and sampled from the stack using a 0.4 cm (5/32") inner diameter, 1 m long fluorinated ethylene propylene (FEP) line that extended approximately 30 cm into the stack. Two polytetrafluoroethylene (PTFE) filters (1 micron pore size) were mounted in series to remove aerosol particles and were changed after every burn. The

10 sampling line contained a restriction consisting of a short section of 0.16 cm (1/16") inner diameter tubing installed in between the stack and the filters that lowered the pressure from ∼900 hPa to ∼600 hPa in order to reduce the relative humidity. The residence time in the sampling line was less than 1 s. Additionally, a glyoxal source consisting of a bubbler containing a 40 wt% solution of glyoxal in water was used to periodically add glyoxal to the instrument above both the restriction and the filters to determine any potential losses of glyoxal on the filters.

### 2.2.2 OP-FTIR

The OP-FTIR measured carbon monoxide (CO), carbon dioxide ($CO_2$), methane ($CH_4$), and formaldehyde (HCHO), as well as a variety of other species (Stockwell et al., 2014; Selimovic et al., 2018). The OP-FTIR was mounted on the platform and measured across the diameter of the stack, with a time resolution of ∼0.73 Hz. Reference spectra were taken from both the High-resolution Transmission (HITRAN) spectral database and spectra previously recorded at Pacific Northwest National

Laboratory (PNNL). The collected IR spectra were then fit using the reference spectra to determine the mixing ratios of the various species (Griffith, 1996; Griffith et al., 2012).

### 2.2.3 PTR-ToF

The PTR-ToF was used to measure VOCs with a proton affinity greater than that of water, including formaldehyde, 2,3-butanedione, 2,3-pentanedione, and several other carbonyl species, with a time resolution of 1 Hz (Yuan et al., 2016; Koss

et al., 2018). For some species, such as formaldehyde and acetaldehyde, calibration factors were determined via the addition of standards, but for other compounds such as 2,3-butanedione the calibration factors were calculated using the method of Sekimoto et al. (2017). PTR instruments generally cannot detect glyoxal since the majority of the glyoxal molecules fragment following protonation to make formaldehyde (Stönner et al., 2017), but glyoxal concentrations at the FSL were high enough for the PTR-ToF to observe some glyoxal, although the agreement with ACES was poor (Koss et al., 2018). Additionally, the

detection of methylglyoxal by PTR suffers from an interference from propenoic (acrylic) acid, which has the same formula ($C_3H_4O_2$) and therefore the same exact mass as methylglyoxal, but does not absorb in the visible. Glycolaldehyde has a similar interference from acetic acid, while 2,3-pentanedione has an interference from methyl methacrylate. The other carbonyls discussed in this work (e.g. 2,3-butanedione) generally are not affected by species with the same masses. While the PTR-ToF





had an inlet on the platform, the instrument itself was not mounted on the platform and instead sampled through a 16 m heated transfer line with a residence time of roughly 1 second. Data from the OP-FTIR and ACES are available for all 75 stack burns, but due to different sampling strategies PTR-ToF data are only available for 58 burns.

## 2.3 Data analysis

Fire integrated emission ratios relative to CO (ER, ppbv glyoxal or methylglyoxal per ppmv CO) were calculated using

$$ER = \frac{\Delta X}{\Delta CO} = \frac{\int_{t_{start}}^{t_{stop}} (X_{fire} - X_{bkgd})\mathrm{d}t}{\int_{t_{start}}^{t_{stop}} (CO_{fire} - CO_{bkgd})\mathrm{d}t} \tag{2}$$

where $\Delta CO$ and $\Delta X$ are the background corrected, fire-integrated mixing ratios of CO and the species of interest, X. Fire integrated emission factors (EF, grams of compound X emitted per kg of fuel burned on a dry mass basis) were calculated using the carbon mass balance method by the following equation

$$EF = 1000 \times F_C \times \frac{MM_X}{AM_C} \times \frac{\frac{\Delta X}{\Delta CO}}{\sum_{i=1}^{n}(NC_i \times \frac{\Delta C_i}{\Delta CO})} \tag{3}$$

where $F_C$ is the mass fraction of carbon in the fuel, $MM_X$ is the molecular mass of species X, $AM_C$ is the atomic mass of carbon, $\Delta X/\Delta CO$ is the emission ratio relative to CO for species X, $NC_i$ is the number of carbon atoms in a given species $i$, and $\Delta C_i/\Delta CO$ is the emission ratio relative to CO for that species. For the 58 burns where the PTR-ToF was sampling from the stack, the total carbon mass was calculated using either only OP-FTIR data or by combining the data from both the

OP-FTIR and the PTR-ToF. The addition of the VOCs measured by the PTR-ToF decreased the glyoxal emission factors by only 3% on average. This is consistent with previous results from the FSL (Stockwell et al., 2015) and with past field studies (e.g. Andreae and Merlet, 2001; McMeeking et al., 2009; Akagi et al., 2011), which have found that $CO_2$, CO, and methane generally make up at least 95% of the total emitted carbon mass. We report EFs based on the combined datasets when PTR-ToF data were available and just OP-FTIR data when PTR-ToF data were not available.

The glyoxal to formaldehyde ratio ($R_{GF}$, moles of glyoxal per moles of formaldehyde) was calculated using

$$R_{GF} = \frac{\Delta Glyoxal}{\Delta Formaldehyde} \tag{4}$$

where $\Delta Glyoxal$ and $\Delta Formaldehyde$ are the background corrected, fire-integrated concentrations of those two species. $R_{GF}$ was calculated using formaldehyde from either the PTR-ToF and the OP-FTIR, but since the two instruments generally agreed well and since the OP-FTIR sampled more burns than the PTR-ToF, unless otherwise stated, all $R_{GF}$ values discussed in the

text used OP-FTIR formaldehyde data.

The modified combustion efficiency, MCE, was calculated using

$$MCE = \frac{\Delta CO_2}{\Delta CO_2 + \Delta CO} \tag{5}$$




MCE values can be calculated either as a fire integrated value, where the integrals of $CO_2$ and $CO$ over the course of the fire are used in equation 5, or as an instantaneous value. Unless otherwise noted, all MCE values here are fire integrated. A higher MCE indicates a greater proportion of flaming during the fire, with a value of 0.9 indicating that the fire was roughly half flaming and half smoldering (Akagi et al., 2011). Fuel moisture content, defined as

$$5 \quad Moisture\ content = \frac{wet\ weight - dry\ weight}{dry\ weight} \qquad (6)$$

and fuel elemental composition were also measured for all 75 burns. Moisture contents are given in the supplement (Table S1) and elemental compositions can be found in Selimovic et al. (2018).

## 3   Results

### 3.1   Carbonyl filter transmission

Previous work has shown that glyoxal loss to filters and any aerosol particles collected on the filters is low (Thalman and Volkamer, 2010; Washenfelder et al., 2011). Glyoxal uptake onto aerosol particles is driven by liquid water content (Kroll et al., 2005; Volkamer et al., 2009; Nakao et al., 2012). Biomass burning particles generally are not very hygroscopic (Yokelson et al., 2009; Akagi et al., 2011; Kreidenweis and Asa-Awuku, 2014), and during the 2016 campaign, the ambient relative humidity in the burn chamber was low (25-40%). Additionally, the inlet restriction before the filter further reduced the relative humidity.

Flow through the filter holder was fast (10 liters per minute) to keep the residence time (~0.3 s) to a minimum. However, the aerosol loadings during these experiments were high and resulted in large accumulations of mass on the filters, even for short burns. A fresh filter was used for each burn, but it is possible that the buildup of material on the filters caused losses of glyoxal and methylglyoxal.

    Prior to the deployment to the FSL, filter transmission tests were conducted in Boulder, CO by burning dried pine needles

and branches in a small wood burning stove and then adding glyoxal to the inlet using the bubbler. The data from one of these tests are shown in Fig. 2. The filter was not changed during this experiment, and glyoxal was added before and after each fire to assess potential losses. No glyoxal losses to the filter were observed during these tests.

    Unfortunately, during the FIREX campaign, the bubbler output frequently was unstable, even over short (20 min.) timescales. During times when the bubbler was reasonably stable, the maximum observed loss was only 10%, but the instabilities in the

bubbler output made it difficult to fully constrain this number. Since we did not observe losses during the tests prior to FIREX and given the uncertainty in the transmission measurements made during FIREX, we have not corrected our data for filter loss, and note that our glyoxal emissions might be up to 10% low. Methylglyoxal is even less reactive than glyoxal with respect to aerosol uptake (Kroll et al., 2005), so any loss of methylglyoxal to the filters should be smaller.



### 3.2 Glyoxal emissions

#### 3.2.1 Glyoxal emission ratios and factors

Glyoxal emission ratios and factors for all 75 burns are shown graphically in Panels A and B of Fig. 3, and values can be found in Supplemental Table S2. Burns from the 33 fuel types have been grouped into eight general categories: chaparrals;

realistic coniferous mixes; separate canopy, litter, duff, and rotten logs from coniferous ecosystems; artificial; and other. Duff is organic material that is denser than litter and has undergone more decomposition. The artificial fuels were untreated lumber and excelsior (wood wool), fuels that are unlikely to be major components of biomass burning. We use the term "artificial" in the sense that these fuels have been processed to some degree, and the biomass is not in its natural state. The "other" category consists of fuels that do not fall into one of the previous seven categories, and includes several important fuels such as peat,

rice straw, and yak dung. Bar graphs of the average emission ratios and factors for the first five groups and select other fuels are shown in Supplemental Figure S1.

Glyoxal emission ratios for realistic mix burns averaged $1.71\pm0.22$ ppbv glyoxal/ppmv CO, nearly a factor of 4 lower than the emission ratio used in the global glyoxal budget by Fu et al. (2008). The other categories have similarly low average emission ratios, and the highest emission ratio, 3.67 ppbv glyoxal/ppmv CO from burning rice straw, was 40% lower than the

15 value from Fu et al. (2008). However, the emission ratios measured here are consistent with the glyoxal enhancements from aircraft intercepts of fresh (<2 hr old) biomass burning plumes, which averaged $1.6\pm0.9$ ppbv glyoxal/ppmv CO (Zarzana et al., 2017). Glyoxal emission factors from other laboratory experiments have been reported by McDonald et al. (2000) and Hays et al. (2002), but only two of the fuels used in those studies, ponderosa pine and loblolly pine, overlapped with fuels used here. Hays et al. (2002) burned fresh ponderosa pine needles, and reported emission factors 5 times higher than our fresh

ponderosa pine canopy emission factor. McDonald et al. (2000) averaged emissions from both ponderosa and pinion pine burns, and report a value that is roughly a factor of 2 higher than ours. Emission factors from burning dry loblolly pine needles were reported by Hays et al. (2002), and are over 12 times higher than the emission factor for our loblolly pine needle litter burns.

These discrepancies in emission factors between the two laboratory studies and the burns conducted at the FSL could

be due to systematic differences in the MCE between the two studies (Christian et al., 2003; Yokelson et al., 2008, 2013; Stockwell et al., 2014). However, due to differences in the ratio of glyoxal to other carbonyls such as formaldehyde (see §3.2.2), a more likely explanation is that the method used by the two previous laboratory studies for detecting carbonyls suffers from interferences. In those studies, carbonyls were detected through derivatization followed by separation using high performance liquid chromatography and detection by ultraviolet absorption measurements. It is now known that measurements

of formaldehyde using this method have interferences from unrelated species such as $NO_2$ that react with the derivatizing agent to form products with similar retention times and absorbances (Karst et al., 1993). ACES also measures $NO_2$, and the $NO_2$ concentrations were comparable to those of formaldehyde for many burns. The species that could cause interferences for glyoxal are not known, but given the complexity of fire emissions and the lack of specificity for the derivatization technique, glyoxal measurement from fires using that technique should be treated with caution. More recent work detects the derivatized





product using electrospray ionization coupled to tandem mass spectrometry (e.g. Kampf et al., 2011), which should provide a greater degree of specificity. The optical method used here relies on the unique differential structure in the visible absorption cross section of glyoxal, reducing the potential for interferences.

Examining emissions from individual fuels, peat had the lowest emission ratio by a factor of 5, while rice straw and bear

grass had the highest, consistent with past results for emissions of larger oxygenated aliphatic compounds (Hatch et al., 2015). For the conifer-derived fuels, the duff burns had the lowest emission ratios, while the canopy and realistic mix burns were the highest. Emission ratios depended more on the fuel component (e.g. canopy or duff) than on the dominant tree species, and generally were consistent within each fuel component group. Chaparral emission ratios were low and in between the duff and litter emission ratios. For the emission factors (grams of glyoxal emitted per kg of fuel burned), peat again had the lowest value,

followed by the chaparrals. Duff and litter were again lower than the canopy and realistic mix burns, but the emission factors for these four groups were much closer than the emission ratios. Fuel component again mattered more than species, but there was more variability in the emission factors than in the emission ratios, especially for the canopy burns.

We examined the relationship between EF and either MCE or fuel moisture content to see if this could explain the variability in the observed EFs. Emission factors as a function of MCE and moisture content are shown in Panels A and B of Fig. 4.

Generally, burns with higher MCEs had lower glyoxal emissions. This is unsurprising, as a higher MCE means that a greater fraction of the carbon in the fuel was converted to $CO_2$. Duff and peat did not follow this trend, and despite having MCEs below 0.9 generally had low glyoxal emission factors. Both duff and peat have undergone some amount of decomposition, and for the duff burns this results in a unique VOC emission profile (Sekimoto et al., 2018), so it is not surprising that these two fuels behave differently. However, of the other four main groups (chaparrals, realistic mixes, canopy, and litter), only the

canopy burns covered a wide range of MCE values, and those burns drive most of the observed trend in emission factors. There appears to be a higher correlation between the fuel moisture content and the glyoxal emission factor, with the wetter fuels having higher glyoxal emissions. Peat is again an outlier, with low emissions despite a high moisture content. However, the canopy burns again were the only group with a large range of moisture content values. Additionally, moisture content and MCE generally were inversely correlated, making it difficult to determine which parameter had the greater effect on emission

factor.

Within certain fuel groups, some of the variability in the emission factors did appear to be driven by differences in the moisture content and MCE. The canopy burns of Engelmann spruce and subalpine fir with the highest emission factors also had moisture contents higher and MCE values lower than the other burns of that material. For most of the other fuel groups, the moisture content within the group did not vary significantly, making it difficult to fully constrain the relationship between

glyoxal emissions and moisture content. Additionally, for some of the burns, there was significant variability in the emission factors despite similar conditions. For example, the two ponderosa pine litter burns were both dry (moisture contents of 0.11 and 0.07) and had similar MCEs, but the emission factors differed by a factor of 3. While moisture content and MCE can affect emissions, clearly there are other factors that also play a role.

Multiple burns of chaparral and coniferous fuels were conducted in 2016, allowing for some investigation of the variability

in emissions for those fuels. However, there were several important fuels that were only burned once, such as peat and rice



straw. During El Niño years, peat fires can emit almost as much non-methane organic carbon as all other biomass burning combined, and can negatively impact local-regional air quality (Akagi et al., 2011; Stockwell et al., 2016b). Crop residue burning is also significant on a global scale, and can strongly impact local-regional air quality, and crop residue may be used as biofuel (Yevich and Logan, 2003; Akagi et al., 2011; Stavrakou et al., 2016). Since only one burn each of peat and rice

straw were conducted in 2016, it is difficult to assess the effect of fire to fire variability and fuel differences (e.g. peat from different regions) on the glyoxal emission factors. However, while glyoxal emissions have rarely been measured, emissions of other small carbonyls from peat and various crop residues have been measured in laboratory studies, such as the fourth Fire Lab at Missoula Experiment (FLAME-4) conducted at the FSL in 2012 (Stockwell et al., 2015), and field projects, such as 2015 Nepal Ambient Monitoring and Source Testing Experiment (NAMaSTE) (Stockwell et al., 2016a) and a 2015 study

conducted in the fall of 2015 in Indonesia (Stockwell et al., 2016b). These studies measured carbonyls such as formaldehyde and glycolaldehyde using the same techniques and sometimes the same instruments as the 2016 study, and emission factors from previous work for the small carbonyls were within a factor of 2 of the emission factors measured in 2016 (Koss et al., 2018; Selimovic et al., 2018). Since emissions of glyoxal are generally very well correlated with the emissions of these other small carbonyls (see §3.2.3), particularly formaldehyde, the good agreement with the previous work gives us confidence that

our results from single burns of peat and rice straw are broadly representative of emissions from those fuels.

### 3.2.2  Glyoxal to formaldehyde ratio, $R_{GF}$

Glyoxal to formaldehyde ratios, $R_{GF}$, for all the burns using OP-FTIR formaldehyde are shown in Panel C of Fig. 3, and bar graphs of $R_{GF}$ for certain fuels are shown in Supplemental Figure S2. $R_{GF}$ values for each fire using either OP-FTIR or PTR-ToF data are available in Supplemental Table S2. Formaldehyde measurements from the two instruments agreed to within 10%

(campaign average) (Koss et al., 2018), and $R_{GF}$ generally was not significantly affected by the choice of instrument. The main exceptions were several of the litter burns and the rice straw burn, where $R_{GF}$ calculated with OP-FTIR data was higher than $R_{GF}$ from PTR-ToF data (e.g. for the rice straw burn $R_{GF}$ calculated using OP-FTIR data is 0.11, compared to an $R_{GF}$ of 0.08 when using PTR-ToF data). $R_{GF}$ across all fuels averaged 0.068±0.018 when using OP-FTIR formaldehyde and 0.060±0.025 when using formaldehyde from the PTR-ToF.

These values are at least an order of magnitude lower than those reported from previous laboratory burns (McDonald et al., 2000; Hays et al., 2002), but are comparable to column measurements by satellites of $R_{GF}$ (0.05-0.08) over regions dominated by biomass burning (Chan Miller et al., 2014; Stavrakou et al., 2016). Zarzana et al. (2017) measured $R_{GF}$ in nighttime plumes that were roughly several hours old, and in daytime plumes less than an hour old. While $R_{GF}$ values in the fresh daytime plumes were comparable to those measured at the FSL (0.06-0.11), $R_{GF}$ values in the nighttime plumes were roughly 40%

lower (0.009-0.04). With the exception of one daytime plume that mostly likely came from burning sugarcane fields, the fuels being burned were not known, so whether the lower nighttime $R_{GF}$ values observed by Zarzana et al. (2017) were due to different chemistry in the fire plumes or different fuel types cannot be determined at this time.

Unlike the glyoxal emission ratios and factors, $R_{GF}$ was consistent across many of the burns, even for unrelated fuels such as chaparral and conifers that had distinct glyoxal emission ratios and factors. The main exceptions were the fuels that had





undergone some form of decomposition, such as duff and peat, which have $R_{GF}$ values 2 to 4 times lower than the others. Given the uniqueness of the duff VOC profiles (Sekimoto et al., 2018), the different $R_{GF}$ for these fuels is not surprising.

Unlike the emission ratios and factors, $R_{GF}$ showed little dependence on moisture content and MCE (Fig. 4c-d). Fig. 4c has an apparent positive correlation between $R_{GF}$ and MCE, but this is driven entirely by the low $R_{GF}$ and MCE values from the

duff and peat burns. For the other burns, $R_{GF}$ showed little dependence on MCE. The only conifer burns that were conducted at low (<0.9) MCE were the duff burns, so it is hard to draw conclusions based on the fire averaged MCE values (see below for discussion of instantaneous MCE values). Duff and peat were outliers in the plot of $R_{GF}$ versus moisture content, but in general no trend was observed between those two parameters. In particular, the canopy burns had a wide range of moisture contents but only a very narrow range of $R_{GF}$ values.

In addition to fire averaged $R_{GF}$ values, we examined the correlation between glyoxal and formaldehyde emissions at each point in the fire. For most of the fuels, glyoxal emissions were well correlated with formaldehyde emissions in real time but were not well correlated with real-time CO measurements. Additionally, there was not a consistent and strong relation between glyoxal emissions and instantaneous MCE. This is shown in Fig. 5. Panels A and B display glyoxal versus CO and formaldehyde respectively for Fire 016, a ponderosa pine litter burn. The markers are colored by the instantaneous MCE.

Glyoxal and formaldehyde are highly correlated ($R^2$=0.94), but the correlations with either CO or MCE are poor ($R^2 \sim 0.3$ for both). Instantaneous $R_{GF}$ was constant over the entire burn, despite the changes in instantaneous MCE. The other burn of ponderosa pine litter (Fire 038, not shown) had a similar fire integrated MCE and fuel moisture content, but had a glyoxal emission factor 3 times higher than the emission factor in Fire 016 (0.189 g glyoxal/kg fuel versus 0.063 g glyoxal/kg fuel). During Fire 016, additional fuel was added several times to increase the length of burn, while no additional fuel was added for

Fire 038. Despite the different glyoxal emission factors and fire behavior, these fires had similar $R_{GF}$ values (0.080 for Fire 038 versus 0.062 for Fire 016).

Panels C and D show the same plots, but for Fire 073, a ponderosa pine rotten log. There are two distinct glyoxal to CO emission ratios, one corresponding to the start of the burn when no flames were present, and the second from the end of the burn when there were flames. For Fire 073, the emission ratios during the non-flaming period at the start and the flaming period

at the end of the burn differed by a factor of 20, but despite this, $R_{GF}$ was constant during the entire duration of the fire and consistent with the ratio from other fuels (0.06 compared to the average of 0.068±0.018). While for Fire 073 it does appear that there is a correlation between instantaneous MCE and glyoxal emission ratio, with the lower MCE corresponding to higher glyoxal emissions, this trend was not observed for many other burns, such as Fire 016, where the highest MCE and emission ratio both occurred at the start of the fire.

### 3.2.3  Correlations with other carbonyls

In addition to formaldehyde, we compared emissions of glyoxal to several other carbonyl species measured by the PTR-ToF: acetaldehyde; acetone; 2,3-butanedione; hydroxyacetone; and glycolaldehyde. The latter two of these species are also measured by the OP-FTIR, but the PTR-ToF data were at the same time resolution as the ACES data, so we chose to use those data here.





There is good overall agreement between the PTR-ToF and the OP-FTIR for these species (Koss et al., 2018), so the results using OP-FTIR data should be similar.

Formaldehyde had the best correlation, with an average $R^2$ of 0.91, followed by acetaldehyde with an $R^2$ of 0.85. For the other carbonyls, $R^2$ values were between 0.75 and 0.79. While glyoxal emissions were only 6-7% of those of formaldehyde, this ratio was higher for the other carbonyls, with glyoxal emissions being roughly 20% of those of acetaldehyde and approximately equal to emissions of 2,3-butanedione and hydroxyacetone. Correlations plots of glyoxal versus four of the other carbonyls for Fire 027, a chamise chaparral fire, are shown in Fig. 6. Glyoxal to formaldehyde plots for the other fires are generally similar, with well correlated and linearly-related emissions for the two species. For acetaldehyde, while many fires resemble Fire 027, in other fires the emissions of glyoxal and acetaldehyde are less well correlated. The other four carbonyl species behave similarly to each other, and the correlations generally decrease as the carbonyl size increases.

## 3.3 Methylglyoxal emissions

### 3.3.1 Spectral retrieval

While the $NO_2$ and glyoxal cross sections are highly structured, the methylglyoxal cross section is not, particularly at the ACES instrument resolution of 1 nm FWHM. This can be seen in Fig. 7a, which shows the absorption cross sections of three of the main absorbers in the ACES retrieval window: $NO_2$; glyoxal; and methylglyoxal. In addition to methylglyoxal, there are several other substituted $\alpha$-dicarbonyls such as 2,3-butanedione and 2,3-pentanedione that have absorption cross sections similar to that of methylglyoxal, albeit with lower magnitudes. The 2,3-butanedione and 2,3-pentanedione cross sections are also shown in Fig. 7a, and the lack of structure in cross sections of the three substituted $\alpha$-dicarbonyls, especially compared to the structure present in the $NO_2$ and glyoxal cross sections, can be clearly seen.

Fig. 7b shows the methylglyoxal, 2,3-butanedione, and 2,3-pentanedione cross sections. While the methylglyoxal cross section has several features between 440 and 450 nm that are not present in the other two cross sections, these features are usually too small to be observed in the measured spectra, except at high concentrations. The fit results from the peak of emissions during Fire 060 (rice straw) are shown in Fig. 8, and at these methylglyoxal concentrations the small features can be resolved, indicating that at least part of the signal attributed to methylglyoxal is indeed from that molecule. However, previous work has shown that the other substituted $\alpha$-dicarbonyls are emitted from biomass burning in amounts comparable to the methylglyoxal emissions we measured at the FSL (Gilman et al., 2015; Stockwell et al., 2015; Koss et al., 2018), and the contribution of these species to the measured extinction needs to be taken into account to properly retrieve the methylglyoxal concentrations.

Other techniques for the measurement of methylglyoxal also suffer from interferences. Methylglyoxal measurements by PTR-ToF are complicated by the presence of an isomer, propenoic (acrylic) acid, which has been measured in fire emissions at the FSL in 2009 using negative-ion proton-transfer chemical-ionization mass spectrometry (Veres et al., 2010) and in 2016 using iodide chemical ionization mass spectrometry ($I^-$ CIMS) (Koss et al., 2018). For the 2016 campaign, the calibration factor for propenoic acid on the $I^-$ CIMS was directly measured by additions of propenoic acid using a liquid calibration unit,




while the methylglyoxal/propenoic acid calibration factor for the PTR-ToF was estimated using the method of Sekimoto et al. (2017). The sum of methylglyoxal and propenoic acid measured by the PTR-ToF was 30% lower than propenoic acid measured by the I⁻ CIMS and 50% lower than the methylglyoxal measured by ACES (even after applying the corrections to the ACES data discussed below), indicating that the PTR-ToF is substantially underestimating the sum of these compounds. However, a

previous study used PTR instruments and CES instruments similar to ACES to measure methylglyoxal either directly injected into a chamber or formed *in situ* by VOC oxidation, and found agreement within 25% (Thalman et al., 2015).

Emissions at the FSL have also been analyzed using two-dimensional gas chromatography-time-of-flight mass spectrometry (Hatch et al., 2015). Unfortunately, methylglyoxal is too sticky to elute on the GC column used for light compounds, and too light for the column used for polar compounds (L. Hatch, personal communication, 2017), so the relative contribution of

methylglyoxal and propenoic acid to the PTR-ToF signal at m/z 73.0284 Th cannot be quantified at this time. However, it is clear that, at least in fresh emissions, both compounds are present in appreciable amounts, and the signal at that mass should be interpreted as the sum of both compounds.

ACES data from the FSL were analyzed in several ways to try to account for the optical interference on the retrieved methylglyoxal concentrations. While 2,3-butanedione emissions are comparable to methylglyoxal emissions, emissions of

larger $\alpha$-dicarbonyls such as 2,3-pentanedione are at least an order of magnitude lower (based on the GC-PTR-ToF results, less than a third of the signal at m/z 101.06 Th, $C_5H_8O_2H^+$, is due to 2,3-pentanedione) (Stockwell et al., 2015; Koss et al., 2018), so the only optical interference that we will consider is that from 2,3-butanedione.

In previous work, a third or fourth order polynomial was included in the fit to account for drift in the instrument zero signal counts (Min et al., 2016), but given the high peak signal at the FSL (several orders of magnitude greater than ambient),

any changes in the background were small relative to the signal from gas phase absorbers. Due to the lack of structure in the methylglyoxal cross section, the DOASIS fitting software tended to assign a large portion of the signal to the polynomial, rather than to methylglyoxal. The polynomial was therefore excluded from the fits. This did not change the retrieved concentrations of the structured absorbers (glyoxal and $NO_2$), but did increase the retrieved methylglyoxal concentrations by roughly 30%.

When running the DOASIS fits without accounting for the other substituted $\alpha$-dicarbonyls, the extinction attributed to

methylglyoxal, $\alpha_{MG}$, is the product of the methylglyoxal cross section, $\sigma_{MG}$ and the apparent methylglyoxal concentration, $N^*_{MG}$. Since 2,3-butanedione is present, the extinction is rather

$$\alpha_{MG} = N^*_{MG}\sigma_{MG} = N_{MG}\sigma_{MG} + N_{BD}\sigma_{BD} \qquad (7)$$

where $N_{BD}$ and $\sigma_{BD}$ are the concentration and absorption cross section of 2,3-butanedione respectively. There are two ways to account for the interference from 2,3-butanedione: include 2,3-butanedione in the DOASIS fits and attempt to simultaneously

retrieve both methylglyoxal and 2,3-butanedione; or only include methylglyoxal in the DOASIS fit and correct the retrieved methylglyoxal using the 2,3-butanedione concentrations measured by the PTR-ToF:

$$N_{MG} = \frac{N^*_{MG}\sigma_{MG} - N_{BD}\sigma_{BD}}{\sigma_{MG}} = N^*_{MG} - N_{BD}R_{BD,MG} \qquad (8)$$

where $R_{BD,MG}$ is the average ratio of the two cross sections in the ACES fit window ($\sim$0.55).



PTR-ToF data were only available for 58 burns, so all further discussion of the methylglyoxal emissions will be limited to results from those fires. All 33 fuel groups are still represented in this subset of fires.

The DOASIS software in principle can simultaneously retrieve the absolute amounts of methylglyoxal and 2,3-butanedione, but this is complicated by the similarities in the two cross sections and their lack of structure, particularly at the low resolution (∼1 nm FWHM) of the ACES instrument. Including 2,3-butanedione in the DOASIS fits lowered the methylglyoxal emission ratios by 41±17%. However, there are many periods when there were rapid fluctuations in the retrieved concentrations of the two species, caused by the similarity between the two cross sections. Additionally, there were numerous periods when either the retrieved methylglyoxal or 2,3-butanedione concentration was negative. These two behaviors do not give us confidence that DOASIS is correctly dividing the measured extinction between methylglyoxal and 2,3-butanedione. While we cannot rule out changes in the ratio of emitted methylglyoxal to 2,3-butanedione as the cause of the variability in the correction, at least part of the variability also appears to be due to fit instabilities.

Using the 2,3-butanedione concentrations measured by the PTR-ToF to correct the methylglyoxal data could be complicated by the presence of other species at the same mass. However, during FIREX, the contribution of different species to the signal at the 2,3-butanedione mass of m/z 87.0441 Th was well characterized by putting a GC column in front of the PTR-ToF, allowing for the separation and quantification of isomeric compounds. 2,3-butanedione contributed 87% of the signal, while methyl acrylate (5%) and several minor, unidentified compounds (8%) made up the balance of the signal. These fractions were consistent across the 9 fires analyzed with this method (Koss et al., 2018). However, the calibration factor necessary to convert the counts measured by the PTR-ToF into 2,3-butanedione concentrations was not measured but rather calculated using the method of Sekimoto et al. (2017) and has an uncertainty of 50%. This method is likely to produce calibration factors that result in an underestimation of the 2,3-butanedione concentration, and thus using those concentrations to correct the methylglyoxal will result in methylglyoxal emissions higher than the actual values.

Using 2,3-butanedione from the PTR-ToF to correct the ACES methylglyoxal did not result in undesirable and unphysical behavior, and reduced the methylglyoxal emission ratios by 17±6% for the 57 non-peat burns, and by 52% for the peat burn (Fire 055), which was the only burn where 2,3-butanedione concentrations were comparable to methylglyoxal concentrations. Due to the issues with simultaneously fitting two diffuse cross sections in DOASIS, we have chosen to fit the ACES data using only the methylglyoxal cross section (in addition to the $NO_2$ and glyoxal cross sections), and then correct the apparent methylglyoxal concentrations using Eq. 8 and 2,3-butanedione concentrations from the PTR-ToF. Due to the uncertainties associated with the calibration factor for 2,3-butanedione, we increased the 2,3-butanedione reported by the PTR-ToF by 50%, so the methylglyoxal emissions we report are lower limits, and have an estimated uncertainty of -50%/+100%. We note that there is still considerable uncertainty in the methylglyoxal emissions, and reducing this uncertainty will require instruments with greater specificity and sensitivity for methylglyoxal.

### 3.3.2 Methylglyoxal emission ratios and factors

Shown in Fig. 9 are emission ratios, emission factors, and the molar ratio of emitted methylglyoxal to glyoxal. Bar graphs of average values for certain fuel groups are also given in Supplemental Figures S1 and S2. Values for the 58 fires where PTR-ToF





data were available are given in Supplemental Table S3. The chaparral burns had some of the lowest methylglyoxal emission factors and ratios, similar to the results for glyoxal. However, litter and duff emitted considerable amounts of methylglyoxal, with the duff burns emitting roughly 50% more methylglyoxal than the canopy and realistic mix burns. This is quite different from the glyoxal results, where duff and littler emitted little glyoxal compared to the canopy burns. As with glyoxal, peat had

5 the lowest methylglyoxal emissions, while rice straw had some of the highest.

Emissions of methylglyoxal from fresh ponderosa pine needles and dead loblolly pine needles have been previously reported by Hays et al. (2002). While that study reported glyoxal emission factors several times higher than ours, the methylglyoxal emission factors are at most only 30% higher than the ones reported here. Emissions for the signal at m/z 73.0284 Th have been reported previously by Stockwell et al. (2015) and Koss et al. (2018). As noted above, the signal at this mass is due

to a combination of methylglyoxal and propenoic acid, with calculated, not measured, calibration factors, so the comparisons between this work and those studies should be treated with caution. Generally, the emission factors from Stockwell et al. (2015) for chaparrals and ponderosa pine are comparable to ours, although we see higher methylglyoxal emissions from rice straw. Our emission factors are higher than those from Koss et al. (2018), with better agreement for the conifers (30% difference) than for the chaparrals and rice straw (factor of 2).

For all the burns, molar emissions of methylglyoxal exceeded those of glyoxal, generally by a factor of 2 and by a factor of 15 for the duff burns. This is consistent with the limited field data, which also found methylglyoxal emissions to be higher than glyoxal emissions (Zarzana et al., 2017), but is in contrast to the results of Hays et al. (2002), who reported glyoxal emissions that were twice as high as methylglyoxal emissions. While the glyoxal and methylglyoxal budgets from Fu et al. (2008) also predict that biomass burning emits more glyoxal than methylglyoxal, this is due to the high glyoxal emissions used, as the

methylglyoxal emissions used in that study are comparable to those observed here.

## 4 Implications

Budgets for glyoxal and methylglyoxal predict that the largest global source for both compounds is VOC oxidation (Fu et al., 2008; Myriokefalitakis et al., 2008; Stavrakou et al., 2009a). However, on local scales emissions of glyoxal and methylglyoxal from biomass burning are expected to dominate over other sources, even with our lower glyoxal emission factors. For example,

during the Southeast Nexus (SENEX) campaign in the summer of 2013 in the southeastern United States, the large regional emissions of isoprene resulted in ambient glyoxal mixing ratios of roughly 100 pptv, ten times lower than what was measured in biomass burning plumes (Kaiser et al., 2015; Zarzana et al., 2017).

The effects of our revised emission factors on the global budgets for these two compounds are harder to quantify. Stavrakou et al. (2016) analyzed emissions from crop residue fires (mainly wheat and maize) in the north China plain measured by

30 the OMI instrument, and were able to model formaldehyde and $NO_2$ columns using literature emission factors for those compounds. The glyoxal column measurements were also compared to the model, and the observed column enhancements were best reproduced using a glyoxal emission factor of 1.12 g glyoxal/kg fuel, over a factor of three higher than our rice straw emission factor (0.34 g glyoxal/kg fuel). Several studies have examined the impact of post-harvest practices on crop burning



emission factors, and found that when the crop residue is piled (mostly commonly in Asia), the fuel tends to smolder for long periods, resulting in lower MCEs and emission factors at the lowest MCEs two to three times higher than those at the highest MCE (Akagi et al., 2011; Inomata et al., 2015; Lasko and Vadrevu, 2018). Our rice straw burn was an open burn, where the fuel was not piled, and had a high MCE (∼0.95), so crop residue burns where the fuel is wetter and piled may have higher

emission factors. However, aircraft intercepts of fresh biomass burning plumes that likely originated from crop residue fires in the southeastern United States have found glyoxal enhancements relative to CO similar to those observed at the FSL (Zarzana et al., 2017), in accordance with the tendency not to pile residues for burning in developed countries (Akagi et al., 2011).

Stavrakou et al. (2016) speculated that some of the glyoxal observed from the satellites could be due to secondary production in the biomass burning plumes. Many of the plumes studied by Zarzana et al. (2017) were emitted at dusk, and two of the

daytime plumes were less than an hour old, limiting any secondary photochemistry leading to glyoxal production in those plumes. While to date there have been no measurements of glyoxal production (or loss) in aged fire plumes, in numerous studies, formaldehyde has been observed to increase relative to CO downwind of fires (Yokelson et al., 2009; Akagi et al., 2012, 2013; Müller et al., 2016). Our measurements of the glyoxal to formaldehyde ratio for fresh emissions are similar to the ratio of total column glyoxal to formaldehyde retrieved by satellites (Chan Miller et al., 2014; Stavrakou et al., 2016), but are

higher than those observed by Zarzana et al. (2017). Glyoxal and formaldehyde have similar lifetimes with respect to photolysis (Volkamer et al., 2005; Röth and Ehhalt, 2015) and oxidation by OH (Feierabend et al., 2008; Burkholder et al., 2015), and if formaldehyde is increasing downwind of fires, then glyoxal must also be increasing if $R_{GF}$ remains roughly constant. Zarzana et al. (2017) observed $R_{GF}$ values 40% lower than the ones we observed at the FSL, so it is possible that the timing of the increases in these two compounds is different. Unfortunately, there have been no measurements of changes in glyoxal as a fire

plume ages, and these measurements are crucial to constraining secondary glyoxal chemistry downwind of fires.

The global glyoxal and methylglyoxal budgets by Fu et al. (2008) predict that oxidation of isoprene by OH is the dominant source of both compounds (∼50% for glyoxal and ∼78% for methylglyoxal). However, since that study, the mechanism and products of the oxidation of isoprene by OH have been examined in much greater detail both theoretically and experimentally (e.g Wennberg et al., 2018, and references therein). Despite this, there is still disagreement in models as to the effect of $NO_x$

on glyoxal yields. The latest version of the Master Chemical Mechanism (MCM v3.3.1) predicts that glyoxal yields will increase as $NO_x$ increases (Jenkin et al., 2015). Two studies examined glyoxal measurements from SENEX using different mechanisms and chemical transport models. Both Li et al. (2016) and Chan Miller et al. (2017) found that the best agreement between the measurements and their respective models came from isoprene oxidation mechanisms where the glyoxal yield decreases with increasing $NO_x$. In particular, the mechanism used by Chan Miller et al. (2017) showed no dependence on $NO_x$

over short (30 minute) timescales. Unfortunately, laboratory measurements of glyoxal and methylglyoxal yields from isoprene oxidation under low $NO_x$ conditions are lacking, and will be required to better constrain the global budgets of both compounds. Additionally, the secondary production of glyoxal and methylglyoxal in fire plumes, and the potential $NO_x$ dependence of that chemistry, has not been measured in either a field or laboratory setting.

Since $R_{GF}$ can be measured from satellites, several studies have examined its utility as a tracer for VOC oxidation. In areas

where isoprene is the main VOC being oxidized, $R_{GF}$ was less than 0.025 (Kaiser et al., 2015), while in areas where aromatics



are the dominant VOCs, $R_{GF}$ is higher (>0.08) (Chan Miller et al., 2016). $R_{GF}$ from fresh biomass burning is the same for many different fuel types and unaffected by parameters such as MCE and fuel moisture content. While the $R_{GF}$ values from fresh biomass burning are distinct from $R_{GF}$ values from isoprene oxidation, further work to determine if $R_{GF}$ remains constant during aging of BB VOC will be an important next step in defining the utility of this metric for investigations of VOC sources
from remote sensing instruments.

## 5   Conclusions

Emissions of glyoxal and methylglyoxal from biomass burning have been determined for a number of different fuels, including peat, rice straw, chaparrals, and numerous conifers. Both compounds were measured using cavity enhanced spectroscopy, which for glyoxal provides a highly sensitive measurement with minimal interferences. The detection of methylglyoxal us-
ing this method suffers from interferences from structurally similar compounds, but due to the high concentrations present, methylglyoxal emissions could be constrained to within a factor of two. Methylglyoxal emissions were higher than glyoxal emissions, and some fuels that emitted little glyoxal emitted large amounts of methylglyoxal. Primary emissions of glyoxal were significantly lower than those reported in previous laboratory work, but were consistent with field measurements in fresh plumes. Glyoxal emissions showed variability between fuel groups, but in nearly all cases were well correlated with emissions
of formaldehyde. The ratio of glyoxal to formaldehyde was consistent at 0.06-0.07 for many of the fuels, with the notable exceptions of duff and peat, which had $R_{GF}$ values at least a factor of 2 lower.

*Data availability.*   Data from all instruments are available at: https://esrl.noaa.gov/csd/groups/csd7/measurements/2016firex/FireLab/DataDownload/.

*Competing interests.*   The authors declare no competing interests.

*Disclaimer.*   TEXT

*Acknowledgements.*   The authors thank Prof. Ryan Thalman (Snow College, UT) for useful discussion. The authors also thank all those who helped organize and participated in the 2016 FIREX intensive, particularly Edward O'Donnell and Maegan Dills for lighting the fires, Ted Christian, Roger Ottmar, David Weise, Mark Cochrane, Kevin Ryan, and Robert Keane for assistance with the fuels, and Shawn Urbanski and Thomas Dzomba for logistical support. Support for V. Selimovic and R. Yokelson was provided by NOAA-CPO grant NA16OAR4310100. A. Koss was supported by funding from the NSF Graduate Fellowship Program. K. Sekimoto acknowledges funding from the Postdoctoral
Fellowships for Research Abroad from Japan Society for the Promotion of Science (JSPS) and a Grant-in-Aid for Young Scientists (B)



(15K16117) from the Ministry of Education, Culture, Sports, Science and Technology of Japan. M. Coggon was supported by a CIRES Visiting Postdoctoral Fellowship. This work was also supported by NOAA's Climate Research and Health of the Atmosphere initiative.

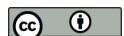



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



**Table 1.** Details of the measurements and instruments used in this work. All three instruments can measure additional species not used in this analysis.

| Instrument | Measured species used in this analysis | Uncertainty | Reference |
|---|---|---|---|
| ACES | glyoxal, methylglyoxal | glyoxal: $\pm 15\%$, methylglyoxal: -50/+100% | Min et al. (2016) |
| OP-FTIR | CO, CO$_2$, CH$_4$, formaldehyde | CO, CO$_2$, CH$_4$: $\pm \sim 2\%$, formaldehyde: $\pm 10\%$ | Stockwell et al. (2014); Selimovic et al. (2018) |
| PTR-ToF | formaldehyde, acetaldehyde, acetone, glycolaldehyde, methylglyoxal, hydroxyacetone, 2,3-butanedione, 2,3-pentanedione | formaldehyde, acetaldehyde, acetone, glycolaldehyde: $\pm 15\%$ All others: $\pm 50\%$ | Yuan et al. (2016); Koss et al. (2018) |





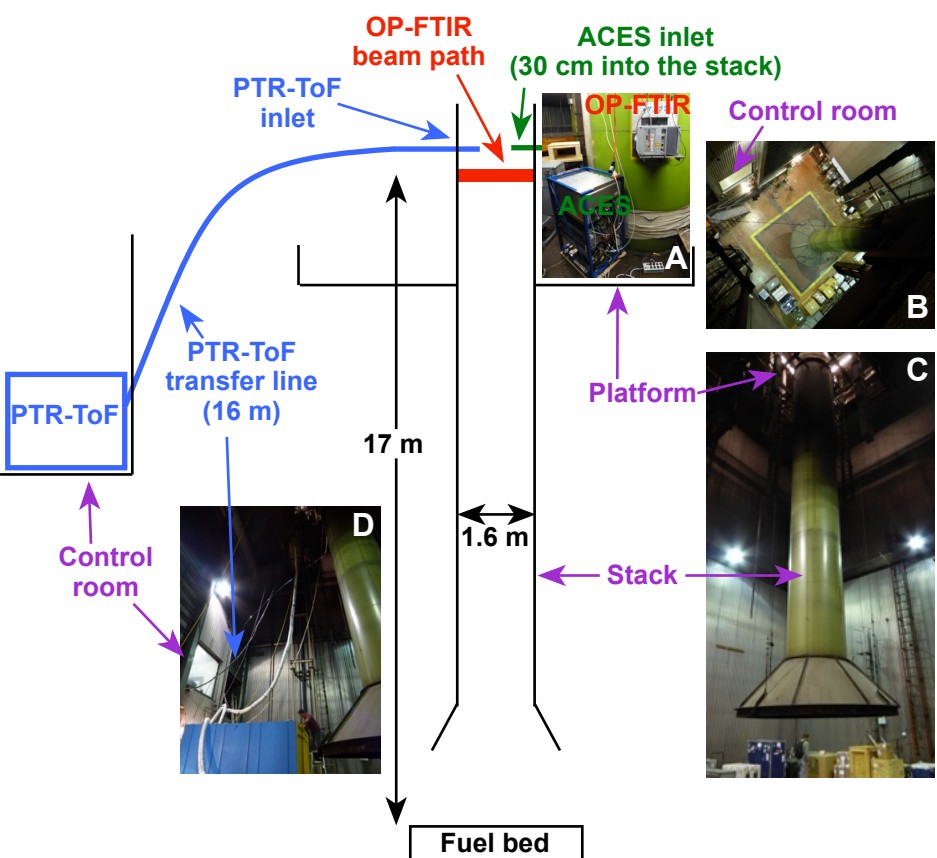

**Figure 1.** Setup of ACES, the OP-FTIR, and the PTR-ToF at the FSL during the 2016 campaign (diagram not to scale). (A) Installation of ACES and the OP-FTIR on the platform. The inlet for ACES was located immediately above the OP-FTIR. (B) View from the platform looking down to the burn chamber floor showing the stack and the window of the control room, where the PTR-ToF was located. (C) View of the stack and platform from the burn chamber floor. (D) View of the stack and control room from the burn chamber floor, showing the PTR-ToF transfer line.





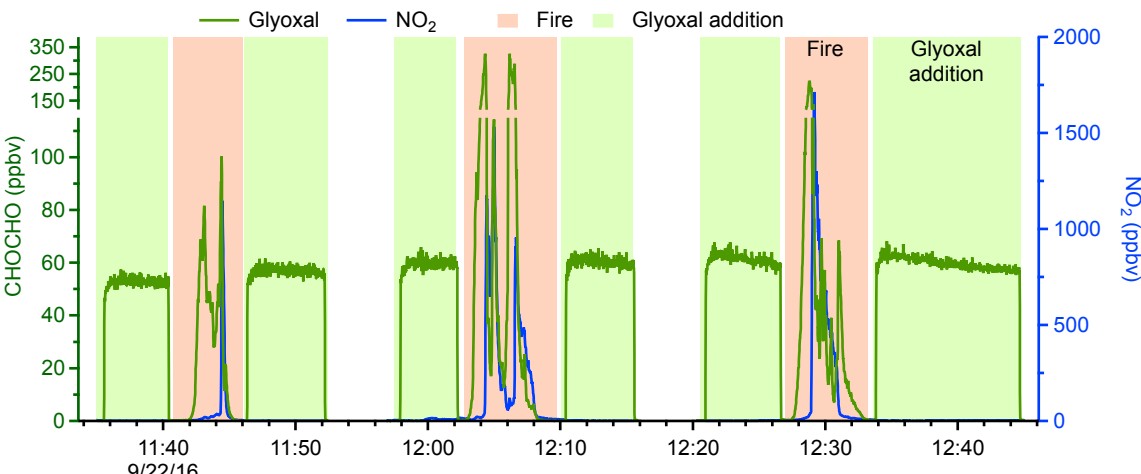

**Figure 2.** Data from a filter transmission experiment conducted in Boulder, CO prior to the FIREX campaign. Shown are the retrieved glyoxal (green) and $NO_2$ (blue) concentrations from the three fire periods (orange shading) and the additions using the bubbler (green shading). The filter was not changed during the experiment.





**Figure 3.** (A) Glyoxal emission ratios in units of ppbv glyoxal/ppmv CO for the different fuel types. The number in the label denotes the number of replicate burns for a given fuel type, while each marker represents the value for an individual fire. (B) Glyoxal emission factors in units of g glyoxal/kg fuel. (C) The glyoxal to formaldehyde ratio, $R_{GF}$, calculated using formaldehyde data from the OP-FTIR. For the first five groups, the average and standard deviation of all the burns in that group are shown. For the last three groups, averages are not provided due to the small number of samples and because the fuels in "Other" are unrelated.





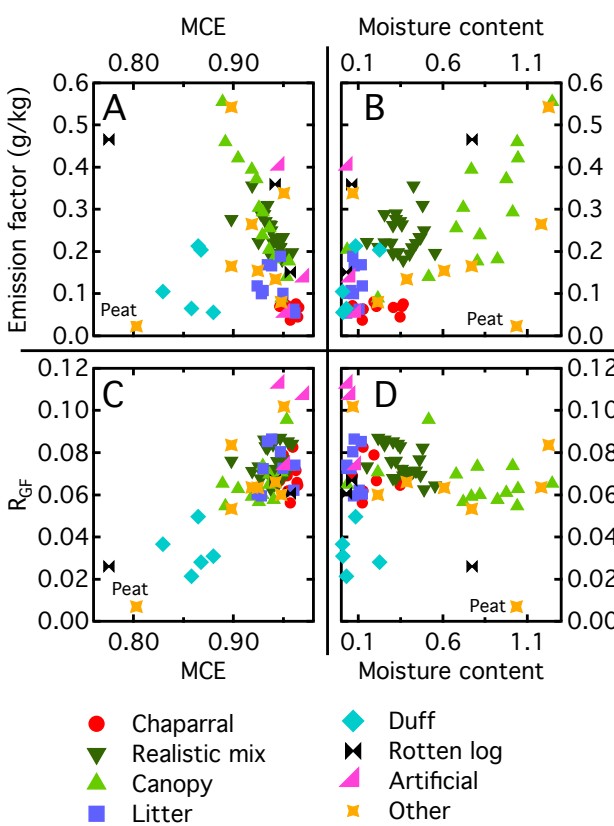

**Figure 4.** Glyoxal emission factors (top) and $R_{GF}$ values (bottom) for each fire as a function of either MCE (left) or fuel moisture content (right).



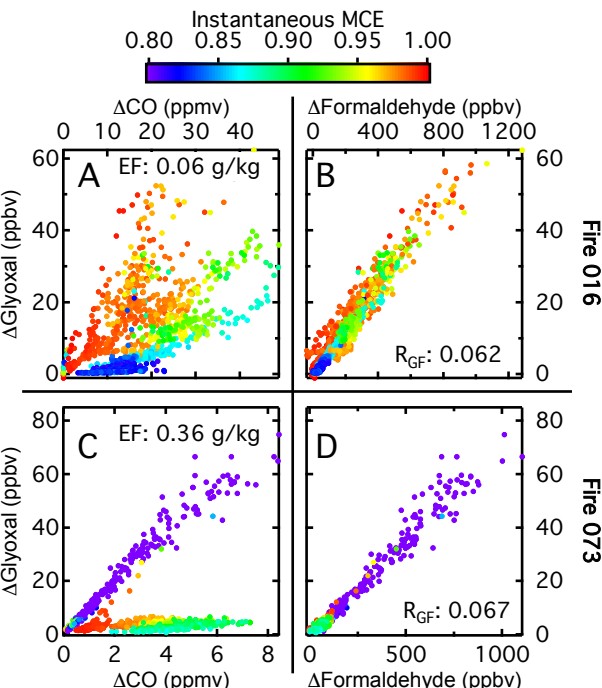

**Figure 5.** (A) Glyoxal as a function of CO for Fire 016 (ponderosa pine litter). (B) Glyoxal as a function of formaldehyde for the same burn. Glyoxal as a function of CO (C) and formaldehyde (D) for Fire 073 (ponderosa pine rotten log). The markers in all plots are colored by the instantaneous MCE.





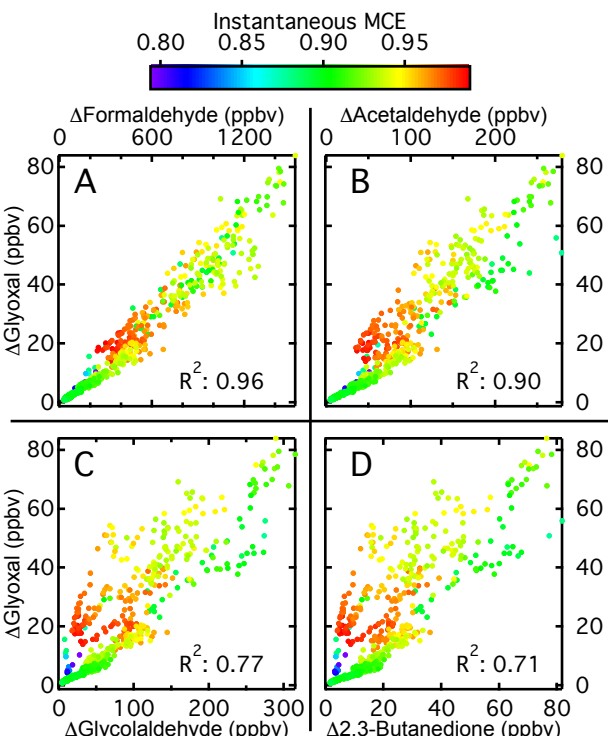

**Figure 6.** Correlation plots for glyoxal relative to four other carbonyls for Fire 027 (chamise chaparral). Shown are the plots for glyoxal versus formaldehyde (A), acetaldehyde (B), glycolaldehyde (C), and 2,3-butanedione (D). The markers are colored by instantaneous MCE. Correlation plots for acetone and hydroxyacetone are similar to the plots for glycolaldehyde and 2,3-butanedione.





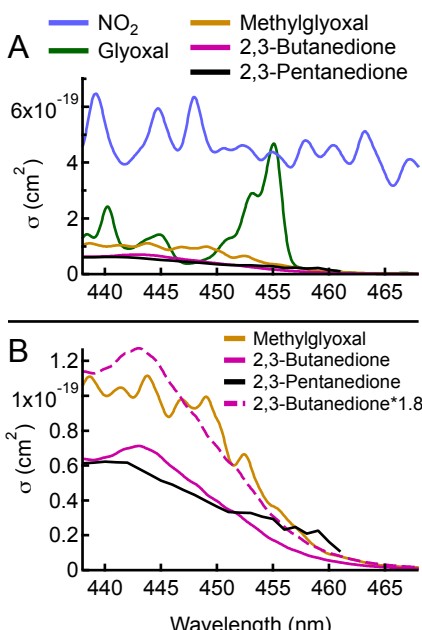

**Figure 7.** (A) Absorption cross sections for $NO_2$ (Vandaele et al., 1998), glyoxal (Volkamer et al., 2005), methylglyoxal (Meller et al., 1991), 2,3-butanedione (Horowitz et al., 2001), and 2,3-pentanedione (Messaadia et al., 2015) in the ACES fit window. (B) Absorption cross sections of methylglyoxal, 2,3-butanedione, and 2,3-pentanedione (solid lines), and the absorption cross section of 2,3-butanedione scaled by a factor of 1.8 (dashed line) to better show the similarity in the shape of that cross section with the methylglyoxal cross section. All the cross sections shown are convolved to the instrument resolution of 1 nm (FWHM).





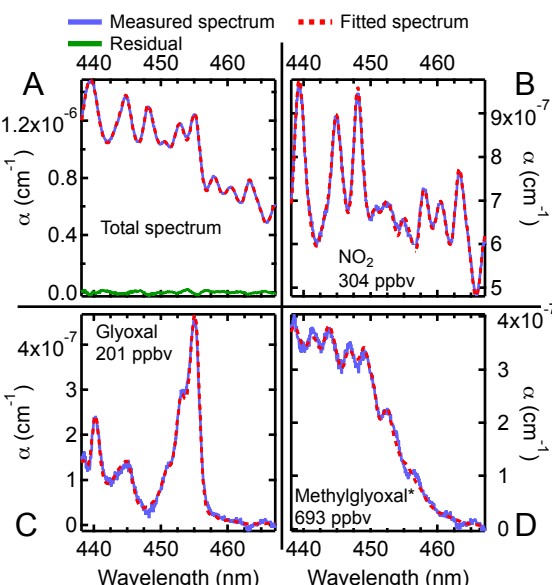

**Figure 8.** Fit results from the peak of emissions from Fire 060 (rice straw). (A) The measured spectrum (blue), the fitted spectrum (red), and the residual (green). Fits for NO$_2$ (B), glyoxal (C), and methylglyoxal (D). At these concentrations, the small features in the methylglyoxal cross section can be resolved. The methylglyoxal concentration given in the figure is the retrieved concentration, and has not been corrected for the interference from 2,3-butanedione.



**Figure 9.** Methylglyoxal emission ratios (A), emission factors (B), and the molar ratio of methylglyoxal to glyoxal (C). Note the split axes for the emission ratio and methylglyoxal to glyoxal plots. Average values for the first five fuel groups and values for certain individual fuels are also shown.