# Peer review of "Primary emissions of glyoxal and methylglyoxal from laboratory measurements of open biomass burning"

_Atmospheric Chemistry and Physics, 2018_

## Referee Comment (RC1) · Anonymous Referee #2 · 16 Jun 2018

General Comments

This paper presents glyoxal and methylglyoxal emissions measured in carefully de-signed and executed experiments from fire lab test burns of a wide variety of biomass types. The results show that glyoxal emissions are lower than suggested by previously measurements (using older methods with more interferences), and methylglyoxal emis-sions are larger than glyoxal by at least a factor of 2. These results appear to explain major discrepancies that have existed for the last several years between field mea-surements and earlier test burn lab measurements. The analytical methods and data workups are described with great clarity, and the goals of the project are well-justified

in the introduction. This work will be of interest to those interested in the atmospheric effects of glyoxal and methylglyoxal or chemical transformation in smoke plumes. Only minor changes are needed before publication.

Specific Comments

My only concern with this paper is an inconsistency in how their results are described. On p. 14 line 23, the authors state that their reported methylglyoxal emissions should be considered lower limits, but then give non-symmetric uncertainty ranges above and below the reported values, which suggests to this reviewer that these values are not really lower limits. Elsewhere in the paper, methylglyoxal emission values are just expressed with the uncertainty ranges, and never described as lower limits. This gives me the feeling that the authors haven't quite made up their minds. If these values really are to be considered lower limits, this information / language should also be included in the abstract, and more consistently described using such language throughout the paper.

Technical Corrections

At a few points in the paper, the authors use the abbreviation "Th" after a molecular mass. What is this abbreviation?

---

## Referee Comment (RC2) · Anonymous Referee #1 · 20 Jul 2018

This paper summarizes the measurements of glyoxal and methylglyoxal made during 75 burns carried out at the Missoula Fire Sciences Laboratory in 2016 as part of the NOAA FIREX campaign. Cavity-enhanced spectroscopy was used to measure both compounds, although the presence of 2,3-butanedione in the smoke may have interfered with the methylglyoxal measurement. The authors find that methylglyoxal emissions were generally 2-3 times larger than glyoxal, in contrast to previous studies which reported higher glyoxal emissions. The authors believe this is due to the interference with other species in the previous glyoxal measurements made by derivatizing the species, using HPLC to separate them, and then measuring the UV absorption. They also find that glyoxal and formaldehyde emissions are highly correlated with each other

and that the ratio of glyoxal to formaldehyde in the fresh emissions is fairly constant for all fuel types.

This is a well written paper on a well performed study focused on an important topic in the chemistry of biomass burning emissions. The uncertainties and possible biases in the measurements are all adequately discussed and their potential impact on the conclusions is noted. The discussion puts the new measurements into the context of previous laboratory and aircraft studies. The figures are clear and the captions are sufficiently detailed.

Honestly, I'm having a hard time finding much to criticize, so I'm going to make a big deal out of some minor typos I found:

P2, L23: Since this is a new paragraph, I'd say "glyoxal and formaldehyde" instead of "these two molecules"

P2, L26: Remove close parenthesis after "∼3 hours"

P15, L4: "duff and littler" should be "duff and litter"

P17, L17: When I go to the website it requests a username and password - you should add how interested researchers can obtain one to the text, or note when the password protection will be removed.

---

## Author Comment (AC1) · 20 Sep 2018

Reviewer #1

*My only concern with this paper is an inconsistency in how their results are described. On p. 14 line 23, the authors state that their reported methylglyoxal emissions should be considered lower limits, but then give non-symmetric uncertainty ranges above and below the reported values, which suggests to this reviewer that these values are not really lower limits. Elsewhere in the paper, methylglyoxal emission values are just expressed with the uncertainty ranges, and never described as lower limits. This gives me the feeling that the authors haven't quite made up their minds. If these values really are to be considered lower limits, this information / language should also be included in the abstract, and more consistently described using such language throughout the paper.*

We agree that this was not clear in the original document and have revised the error calculation and discussion.

The "methylglyoxal" measurement by the ACES instrument, which really is the sum of methylglyoxal and 2,3-butanedione, has an uncertainty of roughly ±30%. The 2,3-butanedione measurement by the PTR-ToF has an uncertainty of ±50%. However, because these two measurements are not independent of each other, we cannot combine the uncertainties in quadrature, so we conservatively have added the absolute uncertainties to arrive at the uncertainty for the difference. This would give an average relative uncertainty of ±70%. However, the method used to determine the PTR-ToF calibration factor for 2,3-butanedione is more likely to give a 2,3-butanedione concentration that is too low, so we increased the 2,3-butanedione concentration by 50% before doing the subtraction. This leads to the asymmetric uncertainty of -30%/+70%.

Since there is a possibility that the methylglyoxal concentrations are lower than the ones we report, we agree that using the phrase "lower limit" is not appropriate and have replaced that phrase with "likely underestimates." We have added this to both the abstract and to the discussion of the methylglyoxal results.

P1 L5: **"Measurements of methylglyoxal using our instrument suffer from spectral interferences from several other species, and the values reported here are likely underestimates, possibly by as much as 70%. Methylglyoxal emissions were 2-3 times higher than glyoxal emissions on a molar basis"**

P14, L21: **"Due to the uncertainties associated with the calibration factor for 2,3-butanedione, we increased the 2,3-butanedione reported by the PTR-ToF by 50%, so the methylglyoxal emissions we report are likely lower than the true values and have an estimated uncertainty of -30%/+70%."**

*At a few points in the paper, the authors use the abbreviation "Th" after a molecular mass. What is this abbreviation?*

"Th" is the symbol for the Thomson, the unit for mass to charge.  For clarity we have removed them.

Reviewer #2

*P2, L23: Since this is a new paragraph, I'd say "glyoxal and formaldehyde" instead of "these two molecules"*
*P2, L26: Remove close parenthesis after "~3 hours"*
*P15, L4: "duff and littler" should be "duff and litter"*

We have corrected these typos.

*P17, L17: When I go to the website it requests a username and password - you should add how interested researchers can obtain one to the text, or note when the password protection will be removed.*

The data are now publicly available.  We have noted this in the text.